# A Novel Laser-Based Zebrafish Model for Studying Traumatic Brain Injury and Its Molecular Targets

**DOI:** 10.3390/pharmaceutics14081751

**Published:** 2022-08-22

**Authors:** Maria A. Tikhonova, Nikolai A. Maslov, Alim A. Bashirzade, Eugenyi V. Nehoroshev, Vladislav Y. Babchenko, Nadezhda D. Chizhova, Elena O. Tsibulskaya, Anna A. Akopyan, Evgeniya V. Markova, Yi-Ling Yang, Kwok-Tung Lu, Allan V. Kalueff, Lyubomir I. Aftanas, Tamara G. Amstislavskaya

**Affiliations:** 1V. Zelman Institute for Medicine and Psychology, Novosibirsk State University, 630090 Novosibirsk, Russia; 2Scientific Research Institute of Neurosciences and Medicine (SRINM), 630117 Novosibirsk, Russia; 3Khristianovich Institute of Theoretical and Applied Mechanics SB RAS, 630090 Novosibirsk, Russia; 4Research Institute of Fundamental and Clinical Immunology (RIFCI), 630099 Novosibirsk, Russia; 5Department of Biochemical Science and Technology, National Chiayi University, Chiayi 600355, Taiwan; 6Department of Life Science, National Taiwan Normal University, Taipei 11677, Taiwan; 7Neurobiology Program, Sirius Science and Technology University, 354349 Sochi, Russia; 8Neuroscience Group, Ural Federal University, 620075 Yekaterinburg, Russia; 9Institute of Translational Biomedicine, St. Petersburg State University, 199034 St. Petersburg, Russia; 10Almazov National Medical Research Centre, 197341 St. Petersburg, Russia; 11Granov Russian Scientific Center of Radiology and Surgical Technologies, 1997758 St. Petersburg, Russia

**Keywords:** traumatic brain injury, laser, animal model, zebrafish, neurodegeneration, neurorepair, neuroinflammation, behavior

## Abstract

Traumatic brain injury (TBI) is a major public health problem. Here, we developed a novel model of non-invasive TBI induced by laser irradiation in the telencephalon of adult zebrafish (Danio rerio) and assessed their behavior and neuromorphology to validate the model and evaluate potential targets for neuroreparative treatment. Overall, TBI induced hypolocomotion and anxiety-like behavior in the novel tank test, strikingly recapitulating responses in mammalian TBI models, hence supporting the face validity of our model. NeuN-positive cell staining was markedly reduced one day, but not seven days, after TBI, suggesting increased neuronal damage immediately after the injury, and its fast recovery. The brain-derived neurotrophic factor (Bdnf) level in the brain dropped immediately after the trauma, but fully recovered seven days later. A marker of microglial activation, Iba1, was elevated in the TBI brain, albeit decreasing from Day 3. The levels of hypoxia-inducible factor 1-alpha (Hif1a) increased 30 min after the injury, and recovered by Day 7, further supporting the construct validity of the model. Collectively, these findings suggest that our model of laser-induced brain injury in zebrafish reproduces mild TBI and can be a useful tool for TBI research and preclinical neuroprotective drug screening.

## 1. Introduction

Traumatic brain injury (TBI) affects nearly 60 million people annually, often causing hospitalization, mortality and permanent disability [1]. TBI is sudden damage to the brain (e.g., caused by a bump, blow, jolt, or penetrating injury to the head) that often occurs after road traffic incidents, falls, military conflicts and various sports [2,3,4,5]. Mild TBI is the most common type of neurotrauma [6]. With a growing prevalence worldwide, managing TBI necessitates both novel therapies and in-depth analyses at the molecular, cellular, and behavioral levels. For example, TBI commonly involves complex secondary molecular cascades that trigger neuroinflammation, glutamate toxicity, blood–brain barrier dysfunction, lipid peroxidation, the release of reactive oxygen species (ROS) and various mitochondrial deficits [7,8]. TBI is also clinically important as it can predispose to multiple neurodegenerative disorders, such as the Alzheimer’s and Parkinson’s diseases, and chronic traumatic encephalopathy [9,10,11,12].

Animal models are widely used to study TBI and TBI-associated pathological neurochemical and neurodegenerative processes, as well as to search for novel anti-TBI therapies [13,14]. For example, rodent weight-drop TBI models replicate focal cerebral contusion, as well as diffuse brain injury with axonal damage [15,16,17]. Other common TBI models involve fluid percussion injury and controlled cortical impact injury [18,19]. However, they all involve surgical procedures [20,21], that make reproducing TBI in these models technically challenging.

The zebrafish (*Danio rerio*) is a popular novel model organism for neuroscience research [22,23,24,25], and can be also applied to studying TBI pathobiology. For example, these fish are suitable both for modeling the symptoms of TBI and its pathogenic mechanisms, as well as for studying TBI-related molecular pathologies, thereby fostering the development of its therapy. There are presently several established zebrafish models of TBI, including (like in rodents) a mechanical blow to the head, accompanied by a peak of the inflammatory response on day 3, and neuroregeneration on day 21 post-insult [26]. Capitalizing on this powerful model species further, here we develop a novel non-invasive method of laser-induced TBI in adult zebrafish.

Lasers have become indispensable tools in biomedicine due to their precision and the ability to produce high density energy flows to cause a precisely localized cell damage for various biological studies [27]. In zebrafish, larvae are an effective model for laser experiments to assess functional relationships between segmental hindbrain neurons [28], visuomotor behavior [29], regeneration after laser axotomy [30], and spinal cord injury research [31].

The present study aimed to: (1) develop a novel model of non-invasive mild TBI in adult zebrafish; and (2) to elucidate potential mechanisms of neurorepair in this model. For this, we used a mutant line of adult zebrafish (*casper*) with transparent skin, to minimize unwanted skin and scull damage by a laser. We also assessed behavioral responses in zebrafish in this TBI model, accompanied by neuromorphological analyses, to further validate the model and evaluate several potential molecular targets for neuroreparative treatment. Here, we focused on core processes of acute period of mild TBI, namely, neurorepair and neuroinflammation. We chose well-validated biomarkers of neuronal injury and repair (NeuN and Bdnf) and neuroinflammation (marker of microglia activation Iba1). In addition, we assessed Hif1a. The latter attracted our attention as a result of its wide study in TBI models and dual effects, promoting both neuroprotection and aggravation of secondary TBI.

## 2. Materials and Methods

### 2.1. Animals

Adult six-month-old experimentally naïve zebrafish of transparent skin mutant *casper* strain were obtained from a local distributor (Individual entrepreneur V.A. Fedoreev, Novosibirsk, Russia). Fish were kept in groups of 500 in 100-L home tanks at pH 7.2–7.4 and maintained at 25–27 °C with a light/dark cycle of 14/10 h. The illumination in the holding room (920 lux) was provided by 18W fluorescent light tubes. Male and female zebrafish were divided into sham-operated (Sham) and laser-injured (TBI) groups (24–36 fish per each group). Each group (Sham and TBI) was tested once on Day 0, 1, 3, 5, or 7 of the study. The mortality rate was 0%. All experimental procedures were carried out in accordance with the NIH Guide for the Care and Use of Laboratory Animals and guidelines established by the Animal Ethics Committee at Scientific Research Institute of Neurosciences and Medicine (Novosibirsk, Russia).

### 2.2. Laser Set-Up

TBI was produced using a laser set-up with a precise targeting system (Figure 1). It consists of a UI-3860CP-C-HQ video camera (IDS Imaging Development Systems, Germany), objective lens with a focal length of 50 mm and a 50-mm extension tube, generating images with a 4-μm resolution. Laser diode with a 405-nm wavelength and 500-mW optical power was used for fish irradiation. The laser condenser lens allows the laser beam to focus into a 0.05-mm spot in diameter. The laser was placed on a mounting mechanism that allows the focus position to adjust, as shown in Figure 1. Control unit sets laser power level in a 1–99% range, with a 1% increment, using a 40-kHz pulse-width modulation, and up to a 2.5-s sample irradiation time, with a 0.01-s increment. The focal spot position on the sample was adjusted using planar translation stage. The video camera sensor was protected from scattered laser light using colored glass optical filter placed in front of the camera lens (Figure 1).

### 2.3. TBI Procedure

Before placement into the laser set-up described above, zebrafish were individually anaesthetized using 100 ppm clove oil, and then fixed in a damp rayon sponge soaked in 100 ppm clove oil. The laser was turned on at a power of 5% of the maximum. This exposure does not cause any damage to the sample but allows control of the focal spot position by observing laser-induced autofluorescence of the brain tissue using a video camera; laser radiation itself is blocked by a protection filter. Using adjustment mechanisms, laser radiation was focused on the target—zebrafish telencephalon. The transparent skin and skull of the fish head enabled laser radiation to be focused on deep tissue layers (the brain) without damaging the surface of the fish skull.

The laser was then switched on at a given power, and for a specific duration. The result of exposure can also be observed visually, by altered color of the affected area, while the color of outer intact tissues remained unaltered (Figure 2). Our approach allows us to localize injury and to vary the damage to nervous tissue by altering the laser power or duration of exposure to the laser irradiation (Figure 2). In our pilot experiments, we established optimal irradiation parameters for the present study. Briefly, the focal spot (0.5 × 0.7 mm) was comparable to the size of the zebrafish telencephalon. The 2.5 s irradiation time was selected to allow heat to dissipate into the tissue and produce more diffuse damage, as compared to shorter impact. With this duration, 35% laser power (175 mW) caused visible damage to the underlying tissues without penetrating the skin surface (further increase in laser power, in contrast, could damage the skin). Upon the parameters used in the present study, we observed blanching of the whole telencephalon and hemorrhages in the olfactory bulbs and optic tectum after the lesion (Figure 2A).

Sham-operated zebrafish were subjected to the same manipulations and procedures. However, instead of being hit by an increased laser power, a harmless 5% laser power was used. The TBI and Sham zebrafish were allowed to recover following injury in 3-L home tanks filled with conditioned tank water.

### 2.4. Behavioral Testing

All behavioral tests were performed in zebrafish during the light cycle between 7:00 AM and 7:00 PM in 30 min (Day 0), one day (Day 1), three days (Day 3), five days (Day 5), and seven days (Day 7) after TBI. To minimize olfactory signals, the water in a testing tank was changed after each fish.

The novel tank test was chosen here as the most sensitive test to assess both locomotion and anxiety in zebrafish [32]. The novel tank test apparatus consisted of a 1.5-L rectangular tank maximally filled with conditioned water and virtually divided into two equal horizontal areas (bottom and top). Once zebrafish were relocated to novel tanks, swimming behavior was recorded using the video-tracking system over a 10-min period. The behavioral measures were automatically registered at a rate of 24 frames/s. EthoVision XT11.5 software (Noldus IT, Wageningen, The Netherlands) was used for behavioral phenotyping. We assessed locomotor activity of the zebrafish by measuring the total distance traveled, freezing frequency and duration, mean velocity. Time spent in the top zone and the latency to enter the top zone were used as well-validated indices of zebrafish anxiety-like states in this test [32].

### 2.5. Cortisol Assay

Samples for cortisol assay were taken from Sham and TBI 50-fish cohorts on Day 0 (30 min after the injury), Days 1, 3, 5, 7 (*n* = 10 per each group) after TBI. In addition, intact zebrafish (*n* = 10) were included into the experimental design, as an additional intact control group. Netted from their home tanks, fish were individually placed in ice-cold water for 1 min and euthanized by decapitation, following the cessation of their opercular movements. Individual body samples were weighted, frozen in liquid nitrogen and stored at −70 °C. Whole-body cortisol was extracted with diethyl ether, according to [33]. To quantify cortisol concentrations, ELISA was performed using a human salivary cortisol assay kit (Xema, Moscow, Russia) according to manufacturer’s instructions, and expressed as content normalized by the sample’s respective weight.

### 2.6. Histological and Immunohistochemical Analyses

Zebrafish were culled in clove oil (100 ppm), their brains were rapidly excised, fixed by 4% paraformaldehyde and postfixed in PBS containing 30% sucrose at 4 °C for neuromorphological analysis. Neuromorphological and immunohistochemical analyses were performed on 30-μm-thick cryosections—coronal slices along the dorsal telencephalic area of each zebrafish brain. For histological examination, hematoxylin and eosin or Nissl staining was applied according to standard protocols. The immunohistochemical analysis was performed according to [33], assessing the expression of the neuronal nuclear protein (NeuN), the brain-derived neurotrophic factor (Bdnf), the hypoxia-inducible factor 1-alpha (Hif1a), and microglial marker Iba1 in zebrafish telencephalon on Days 0 (30 min after injury), 1, 3, 5, and 7 after TBI.

We applied a rabbit polyclonal antibody (ab72439,1:200 dilution, Novus Biologicals, USA) as a primary antibody to detect Bdnf; a rabbit polyclonal antibody (NB100-134, 1:100 dilution, Novus Biologicals, Littleton, CO, USA) as a primary antibody to detect Hif1a; a rabbit monoclonal antibody (ab 177487, 1:200 dilution, Abcam, Cambridge, UK) as a primary antibody to detect the neuronal marker NeuN; a rabbit polyclonal antibody (cat. # 019-19741, 1:200 dilution, FUJIFILM Wako Pure Chemical Corporation, Japan) to detect Iba1. A fluorescently labeled (Alexa Fluor 488–conjugated) goat anti-rabbit IgG antibody (ab150077, 1:500 dilution, Abcam, UK) served as the secondary antibodies for Bdnf, Hif1a, Iba1, and NeuN. Fluorescence intensity was measured as a background-corrected optical density (OD) with subtraction of staining signals of the non-immunoreactive regions in the images converted to grayscale. The area of interest was 18191 μm^2^ in the dorsal pallium. Fluorescent images were finally obtained by means of an Axioplan 2 (Carl Zeiss) imaging microscope and then analyzed in Image Pro Plus Software 6.0 (Media Cybernetics, Rockville, MD, USA).

### 2.7. Statistical Analysis

STATISTICA 10.0 software (StatSoft, Tulsa, OK, USA) was used to perform all the statistical analyses. The normality of the data distribution was determined by the Shapiro–Wilk *W* test. Statistical evaluation of data was performed by analysis of variance (ANOVA) followed by Fisher’s LSD post hoc test for normally distributed data or by Kruskal–Wallis *H* test followed by multiple comparisons of mean ranks for all groups in case of non-normal distribution of the data. In some cases (e.g., comparing TBI vs. Sham groups in behavioral testing on each day), paired Student’s *t*-test for normally distributed data or the Mann–Whitney test with Bonferroni correction were used. Spearman test with *rho* correlation coefficient was applied to assess correlation between the parameters. The level of significance was defined as *p* < 0.05 in all experiments reported here.

## 3. Results

### 3.1. Behavioral Effects of Laser-Induced TBI in Zebrafish

Showing similar recovery time (data not shown), all zebrafish groups regained consciousness within 10 min after the injury.

#### 3.1.1. Locomotor Activity

The novel tank test revealed significant differences in the distance traveled between Sham and TBI groups Days 0 (30 min after lesion; MedianSham0 = 2395 cm, MedianTBI0 = 742 cm, Δ = −69%; *p* < 0.01), 1 (MedianSham1 = 1946 cm, MedianTBI1 = 522 cm, Δ = −73%; *p* < 0.05) and 7 (MedianSham7 = 3143 cm, MedianTBI7 = 1704 cm, Δ = −46%; *p* < 0.001). Locomotor activity in the laser-lesioned zebrafish was substantially attenuated on these days. The Kruskal–Wallis *H* test showed significant changes in behavioral response of both Sham (H(4,N = 153) = 23.4, *p* < 0.001) and TBI (H(4,N = 167) = 25.9, *p* < 0.001) groups depending on time passed since the experimental procedure. However, there were no significant differences between the Sham group on Day 0 and the rest Sham groups. A slight decrease in the locomotion was observed on Day 1 and Day 3 while then the locomotor activity augmented on Days 5 and 7 in Sham zebrafish. In laser-lesioned zebrafish, there was also a slight decrease in the locomotion on Day 1 while an increase in the locomotor activity was observed on Days 3, 5, and 7. Although there was a significant increase in the locomotion of TBI group on Day 5 (MedianTBI5 = 2140 cm) compared to Days 0 (MedianTBI0 = 742 cm) and 1 (MedianTBI1 = 522 cm) (Δ = 65%, *p* < 0.05 and Δ = 76%, *p* < 0.001, respectively) and on Day 7 (MedianTBI7 = 1704 cm) compared to Day 1 (MedianTBI1 = 742 cm) (Δ = 69%; *p* < 0.01), the locomotor activity in laser-lesioned zebrafish on Day 7 remained significantly lower than in the respective Sham group (*p* < 0.001). Thus, the locomotion in zebrafish was substantially reduced by laser-induced TBI 30 min, one and seven days after the injury (Figure 3A).

#### 3.1.2. Freezing Frequency

TBI groups demonstrated more freezing on Days 0 (MedianSham0 = 2, MedianTBI0 = 148, Δ = 7300%; *p* < 0.001) and 1 (MedianSham1 = 7, MedianTBI1 = 105, Δ = 1400%; *p* < 0.01) vs. the Sham groups. Significant alterations in the parameter were observed in both Sham (H(4,N = 153) = 13.9, *p* < 0.01) and TBI (H(4,N = 167) = 40.8, *p* < 0.001) groups depending on time passed since the experimental procedure, with less freezing frequency in the Sham groups on Day 5 (MedianSham5 = 1) vs. Day 1 (MedianSham1 = 7) (Δ = −600%, *p* < 0.01). In TBI groups, the index was markedly lower on Days 5 (MedianTBI5 = 6) and 7 (MedianTBI7 = 3.5) vs. Days 0 (MedianTBI0 = 148) (Δ = −2367%, *p* < 0.001 and Δ = −4129%, *p* < 0.001, respectively) and 1 (MedianTBI1 = 105) (Δ = −1650%, *p* < 0.001 and Δ = −2900%, *p* < 0.001, respectively) (Figure 3B). Thus, the freezing frequency in zebrafish was markedly augmented by laser-induced TBI in 30 min and one day after the injury. The frequency of freezing was significantly reduced (vs. Days 0 and 1) starting from Day 5.

#### 3.1.3. Freezing Cumulative Duration

TBI groups demonstrated longer freezing on Days 0 (MedianSham0 = 1.1 s, MedianTBI0 = 139.6 s, Δ = 12822%; *p* < 0.05) and 1 (MedianSham1 = 66.7 s, MedianTBI1 = 322.9 s, Δ = 384%; *p* < 0.05) compared to the Sham groups. Significant alterations in the parameter were observed in both Sham (H(4,N = 153) = 13.8, *p* < 0.01) and TBI (H(4,N = 167) = 23.3, *p* < 0.001) groups depending on time passed since the experimental procedure. In TBI groups, this index was markedly lower on Day 5 (MedianTBI5 = 13.6 s) compared to Day 1 (MedianTBI1 = 322.9 s) (Δ = 2271%, *p* < 0.001) (Figure 3C). Thus, freezing duration here was markedly augmented by laser-induced TBI 30 min and one day after the injury and had returned to control values of Sham groups since Day 3.

#### 3.1.4. Velocity

Significant differences in the velocity between Sham and TBI groups were revealed on Days 0 (MedianSham0 = 3.99 cm/s, MedianTBI0 = 1.24 cm/s, Δ = −69%; *p* < 0.05) and 1 (MedianSham1 = 3.79 cm/s, MedianTBI1 = 0.87 cm/s, Δ = −77%; *p* < 0.001) of the study. Mean velocity in the laser-lesioned zebrafish was lower on these days than in respective Sham groups. Both Sham (H(4,N = 153) = 15.9, *p* < 0.01) and TBI (H(4,N = 167) = 23.9, *p* < 0.001) groups differed significantly in the mean velocity depending on time passed since the experimental procedure. There was a significant increase in the velocity of Sham groups on Days 5 (MedianSham5 = 5.02 cm/s) and 7 (MedianSham7 = 5.33 cm/s) vs. Day 1 (MedianSham1 = 3.79 cm/s) (Δ = 25%, *p* < 0.05 and Δ = 29%, *p* < 0.05, respectively). In TBI groups, the parameter was markedly higher on Day 5 (MedianTBI5 = 3.57 cm/s) vs. Day 0 (MedianTBI0 = 1.24 cm/s) (Δ = 65%, *p* < 0.05) and on Days 3 (MedianTBI3 = 2.4 cm/s), 5 (MedianTBI5 = 3.57 cm/s), and 7 (MedianTBI7 = 3.88 cm/s) vs. Day 1 (MedianTBI1 = 0.87 cm/s) (Δ = 64%, *p* < 0.05, Δ = 76%, *p* < 0.001, and Δ = 78%, *p* < 0.01, respectively; Figure 3D). Thus, the velocity in zebrafish was substantially reduced by laser-induced TBI 30 min and one day after the injury, but recovered after Day 3.

#### 3.1.5. Visiting the Top Zone

Cumulative duration in the top zone was significantly decreased in the laser-lesioned zebrafish compared to Sham groups on Day 0 (30 min after lesion; MedianSham0 = 180.2 s, MedianTBI0 = 53.4 s, Δ = −70%; *p* < 0.05) and Day 1 (MedianSham1 = 88.2 s, MedianTBI1 = 0.3 s, Δ = −99%; *p* < 0.05). According to Kruskal–Wallis *H* test, there were significant changes in the behavioral response of TBI groups (H(4,N = 167) = 13.8, *p* < 0.01) depending on time passed since the experimental procedure. However, there were no significant differences between the Sham groups. In laser-lesioned zebrafish, there was a significant decrease in the duration of staying at the top of a tank on Day 1 (MedianTBI1 = 0.3 s) compared to Day 0 (MedianTBI0 = 53.4 s) (Δ = −16581%, *p* < 0.05) while an increase was observed on Day 5 (MedianTBI5 = 48.4 s) compared to Day 1 (MedianTBI1 = 0.3 s) (Δ = 99%, *p* < 0.05) (Figure 3E). Thus, the duration in top was substantially reduced in zebrafish by laser-induced TBI 30 min and one day after the injury, but fully recovered on Day 5.

#### 3.1.6. The Latency to Enter the Top

The TBI group took significantly longer than the Sham group to enter the top on Day 7 (MedianSham7 = 53.4 s, MedianTBI7 = 451.4 s, Δ = 745%; *p* < 0.05), supporting anxiety-like behavior on Day 7 after the laser-induced TBI (Figure 3F).

### 3.2. Effects of Laser-Induced TBI on Cortisol Levels in Zebrafish

There was a significant time after injury effect (F(4,87) = 14.4, *p* < 0.001) and of the interaction between time and TBI (F(4,87) = 5.8, *p* < 0.001) on the levels of cortisol in fish (Figure 4). The expression of NeuN was markedly higher in TBI vs. the respective Sham groups on Day 0 (30 min after lesion; *p* < 0.001), but dramatically decreased in the laser-lesioned zebrafish on Day 1 (*p* < 0.001 vs. the Sham group on Day 1 or TBI group on Day 0). Cortisol levels were significantly augmented in the TBI group compared to the corresponding Sham group on Day 0 (30 min after the injury; *p* < 0.001). On days 1, 3, 5, and 7 after the injury, the values did not differ significantly between TBI and Sham groups (*p* > 0.05). Notably, the cortisol level in Sham zebrafish did not differ significantly from that in intact animals (Figure 4), suggesting the activation of the adrenocortical system in zebrafish after TBI only on the day of injury.

### 3.3. Effects of Laser-Induced TBI on Neuromorphology, Neuroinflammation, Neuronal Damage and Recovery

Laser irradiation produced clear-cut damage to neural tissue, one day after the injury producing dilated vessels, hemorrhage, and edema (Figure 5) as well as degenerating neurons (Figure 6) in the telencephalon of zebrafish.

Neuronal injury after TBI was assessed using staining the brain sections with antibodies to a specific neuronal marker, NeuN, labeling neurons as NeuN-positive cells. There were significant effects for TBI (F(1,28) = 8.1, *p* < 0.01), time after injury (F(4,28) = 45.4, *p* < 0.001), and their interaction (F(4,28) = 32.3, *p* < 0.001) on the expression of NeuN in zebrafish brain (Figure 7). In general, NeuN labeling was markedly higher in TBI vs. the respective Sham group on Day 0 (30 min after lesion; *p* < 0.001), and dramatically decreased in the laser-lesioned zebrafish on Day 1 (*p* < 0.001 vs. the Sham group on Day 1 or TBI group on Day 0). The NeuN expression gradually increased during the next days of the experiment in the TBI groups, but was significantly lower than in the respective Sham groups on Days 3 (*p* < 0.001) and 5 (*p* < 0.01). On Day 7, NeuN levels recovered in the TBI group up to the Sham group levels. Overall, the expression of neuronal marker NeuN was augmented in the acute period following TBI (30 min after lesion), sharply dropped on Day 1 and then gradually restored up to values of Sham groups by Day 7.

There was a significant effect for TBI (F(1,22) = 14.8, *p* < 0.001), time after injury (F(3,22) = 16.7, *p* < 0.001), and their interaction (F(3,22) = 11.6, *p* < 0.001) for the levels of neurotrophic factor Bdnf (Brain-derived neurotrophic factor) in zebrafish telencephalon (Figure 8). Overall, the expression of Bdnf was much lower in the TBI group vs. the respective Sham group on Day 0 (30 min after the lesion; *p* < 0.001), whereas on Day 1 after the injury, values between the groups did not differ. Bdnf expression increased dramatically on Day 3 post TBI (*p* < 0.001), decreased on Day 5 vs. Day 3 (*p* < 0.001), but remained higher than the respective Sham group (*p* < 0.05). On Day 7, Bdnf levels in the TBI group returned to the levels of the Sham group. In sum, Bdnf significantly decreased in the TBI group on the day of injury (30 min after the lesion) and then sharply rose on Days 3 and 5.

Hif1a is a subunit of the heterodimeric transcription factor hypoxia inducible factor 1 and a key mediator of oxygen homoeostasis [34]. Significant effects of factors of trauma and time, as well as their interaction, were noted (F(1,25) = 53.1, *p* < 0.001; F(4,25) = 4.7, *p* < 0.01; and F(4,25) = 3.8, *p* < 0.05, respectively). The levels of Hif1a were markedly increased in TBI groups on Days 0, 1, 3, and 5 compared to respective Sham groups. On Day 7 after the injury, no significant differences between the TBI and Sham groups were found. Thus, the expression of Hif1a in the telencephalon of laser-lesioned zebrafish increases right on the day of traumatic injury and persists further until Day 7 after the lesion (Figure 9).

Ionized calcium-binding adapter molecule 1 (Iba1) is a microglial marker markedly upregulated by microglia activation [35]. There were significant effects of TBI, time and their interaction (F(1,17) = 129.8, *p* < 0.001; F(4,17) = 8.7, *p* < 0.001; and F(4,17) = 3.2, *p* < 0.05, respectively). The levels of Iba1 were markedly increased in TBI groups during all days of the experiment, compared to respective Sham groups, with a peak on Day 1. However, a gradual relative decrease in the Iba1 expression was seen in the TBI groups compared to acute period of injury (Days 0 and 1 after TBI). In general, elevated Iba1 expression in zebrafish telencephalon by laser-induced TBI occurred on the day of traumatic injury and persisted for at least seven days after the injury (Figure 10).

Finally, there was a strong positive correlation between the expression of Iba1 and Hif1a (*rho* = 0.81, *p* = 0.000004), whereas both parameters correlated negatively with the NeuN expression (*rho* = (−0.55), *p* = 0.006 and *rho* = (−0.395), *p* = 0.028 for Iba1 and Hif1a expression, respectively).

## 4. Discussion

Although modeling the entire spectrum of TBI pathophysiology in animals is impossible given the heterogeneity of the injury, numerous animal models have been developed to understand selected pathophysiological mechanisms of TBI and its consequences, as well as its potential therapeutic targets. Individual aspects of TBI have been reproduced successfully in animal models, including both mammals and zebrafish. Traditionally, an animal model is considered to be valid, if it resembles the human condition in symptomatology (face validity), response to therapeutic interventions or aggravating impacts (predictive validity), pathology mechanisms and theoretical construct about the dysfunctional processes (construct validity) [36].

TBI models usually involve focal, diffuse, or combined injury. For example, diffuse brain injury is characterized by a rapid acceleration–deceleration of the brain, more often without a direct impact to the head. Focal damage is obtained in animals following the impact to a specific area of the exposed dura or directly to the closed skull, causing laceration, contusion and hematoma [37,38]. In zebrafish, the most common TBI models involve penetrating injuries of the brain tissues [39,40], ranging from moderate to severe, including the penetration from the epidermal layer through the blood–brain barrier [41]. However, these models do not adequately recapitulate closed-head TBI.

Recently, two novel models of closed-head TBI have been developed and characterized in adult zebrafish. In one, a classical rodent weight drop model of non-penetrating, diffuse mild TBI was adapted to the adult zebrafish [26]. Another closed-head model of TBI in adult zebrafish involves a targeted pulsed high-intensity focused ultrasound [42]. Both such models reproduce a diffuse type of TBI and affect the whole brain tissue. Due to the small size of zebrafish head, it is challenging to propose a method that allows the choosing of the angle, force, and point of application of the impact for region- specific analyses. To address this need, we present a novel model of focal non-penetrating TBI induced by laser.

The main effects in biological tissues exposed to laser radiation occur due to heating. First, a resonant absorption of a radiation quantum occurs in the vibrational-rotational band of the molecule, which transfers it to an excited state. Then, due to inelastic interactions with the environment of the molecule, its kinetic energy increases and causes a local hyperthermia. Pathological changes occurring in a tissue mainly depend on the radiation power, exposure time, and the depth of penetration into the tissue. During laser-induced heating, biological tissues may show no change (<40 °C), hyperthermia (42–50 °C for short-term impact) or denaturation of proteins and cell necrosis (>60 °C), ablation process (100 °C), carbonization (150 °C) and melting (>300 °C). These effects may happen simultaneously, from carbonization on the surface to hyperthermia several millimeters deeper. Thus, to avoid undesirable thermal damage, selecting the parameters of laser radiation (e.g., period and duration of pulses, pulse energy) becomes important.

Another advantage of laser radiation is the possibility of its distant action, as transparent tissues are not prone to its destructive action. Zebrafish strains vary widely in terms of color and transparency, and include the fish with transparent head areas, thereby enabling laser radiation to damage the brain without impacting surface tissues, such as skin and skull, to provide a controlled non-invasive TBI modeling. Here, we applied laser irradiation produced by the laser diode (power—500 mW, wavelength—405 nm).

Damage to the brain tissue was confirmed by neuromorphological examination of the samples using histological staining. In the acute period of neurotrauma (one day after the injury), injury patterns like dilated vessels, hemorrhage, and edema were observed in zebrafish telencephalon. These findings parallel pathological features in rodent models of mild TBI, and are clinically relevant. Moreover, behavioral alterations produced by laser-induced TBI in zebrafish resemble transient motor and anxiety-related deficits in that models [38]. A similar decrease in motor activity and increased anxiety in the novel tank test have been revealed immediately after injury, 24 and 48 h post-injury in another closed-head model of TBI induced by pulsed high intensity focused ultrasound in adult zebrafish [42].

In the present study, locomotor indices were reduced while anxiety-like behavioral patterns rose in the injured zebrafish on the day of lesion and on the next day and the behavioral alterations had recovered since the Day 3. The TBI group differed in some parameters of locomotion (distance traveled) or anxiety (latency to the first entry to upper zone) from the Sham group on Day 7. Long-term deficits in response to injury are widely reported for mammalian systems [43,44]. Thus, the laser-induced TBI model in zebrafish shares similar symptoms of mild TBI as well as some features of the disease course with mammalian organisms that evidences so-called face or ethological validity of a model [36].

TBI results from an initial insult to the brain (primary injury) followed by a complex secondary response with altered cell proliferation and differentiation, neuronal cell death, mitochondrial dysfunction, multifaceted immune responses, glutamate-induced excitotoxicity and cerebrovascular dysfunction [7]. TBI can also trigger acute and chronic neuroinflammation, oxidative stress, and axonal damage [8] that, in turn, alter synaptogenesis, dendritic remodeling, and neurogenesis in the hippocampal, cortical, and limbic regions [45]. Unlike mammals, zebrafish maintain the remarkable ability to regenerate and repair their brain after an insult, especially due to the large number of neurogenic niches in the brain [46]. In mammals, the proliferative process prevails in the affected area and is glial in nature, while in zebrafish, proliferation takes place mainly in neurogenic zones, from which neuronal precursors then migrate to the injured area. Unlike in mammals, gliosis (increased proliferation of glia in the affected area) is not pronounced in fish and does not have overt pathological effects [40,46], whereas zebrafish radial glia cells have the potential to differentiate into neuroblasts [39,47]. Zebrafish capacity for quick and almost complete functional and morphological neuroregeneration attracts particular attention, and zebrafish TBI models are specifically suited to probe neurorepair mechanisms and discover potential targets for regenerative therapy [26,48,49].

Notably, Bdnf promotes neuronal survival, neuroplasticity, synapto- and neurogenesis, and reduces apoptosis [50,51]—all crucial processes for CNS recovery after TBI. Bdnf also attenuates neuroinflammation, including astrocytosis, microcytosis, and IL-1β expression [52,53]. A remarkable difference exists between zebrafish and rodents in the Bdnf expression, as its upregulation persists around the lesioned area in zebrafish but only in the contralateral side in rodents. Hence, Bdnf is suggested as a core actor providing favorable conditions for complete restoration of the damaged brain tissue [54].

In the present study, the expression of Bdnf was significantly decreased in the TBI group on the day of injury (30 min after the lesion) and then increased with a sharp peak level on the Day 3, returning to sham-operated group levels on Day 7. In rat pups after open-head TBI (controlled cortical impact), there was also a surge in Bdnf on the third day after the lesion and retrieval to the levels of sham animals since the Day 7 [55]. The dynamics are also similar in general to the previous findings about the patterns of the expression of Bdnf mRNA after a stab wound of zebrafish telencephalon, although certain divergence takes place. After the stab wound lesion, a significant increase in Bdnf mRNAs levels occurred one day post lesion compared to control un-lesioned animals, then decreased with time after lesion [54].

In the laser-induced zebrafish TBI model, NeuN expression (labeling mature neurons) was attenuated on Day 1 that is in a good agreement with previous findings in TBI models. In a rat model of TBI induced by free-fall injury, the expression of NeuN was also markedly diminished one day after the trauma [56]. Besides, we found a paradoxical increase in NeuN levels on Day 0 (30 min after TBI) preceding the decline on the next days. NeuN is regarded as a specific neuronal marker of postmitotic neurons. This protein plays a role in neurospecific alternative splicing. The differences in the expression of this protein in cells are associated with both the constitutive characteristics of a neuron and its functional state [57]. Most of the studies of pathological changes in neuronal populations indicate a weakening or disappearance of NeuN immunoreactivity in neurons. We have not found reports about upregulation of NeuN. Moreover, 30 min after injury is too short a period for huge production of NeuN protein de novo that could explain almost a two-fold increase in the NeuN immunofluorescence. NeuN protein is predominantly associated with cell nuclei and, to a lesser extent, with the perinuclear cytoplasm. Most of the intranuclear NeuN is bound to the nuclear matrix [57]. We hypothesize that laser irradiation disturbed that bound, and NeuN was massively released to neuron cytoplasm and thus caused an increase in the OD of NeuN staining. Nevertheless, this issue needs further detailed study.

The levels of NeuN expression had been markedly reduced until Day 7 when the parameter in the TBI group was completely recovered. The time frame is too short for neuronal precursor cells migration from the zones of proliferation to the injury site and for them to differentiate into mature neurons [39]. Moreover, the loss of neuronal NeuN immunoreactivity indicates damage, but cannot prove the neuronal death [57]. Thus, the results suggest neuroprotection or repair of damaged neurons rather than neuroregeneration due to neurogenesis. Nonetheless, potential effects of laser-induced injury on neurogenesis were out of the scope of the present study and would be addressed in further research of this model.

Overall, we consider that Bdnf promotes the survival and functional recovery of mature neurons in the lesioned area as a core event at the acute period of laser-induced mild TBI and recovery after the primary injury in the present model. This notion is also in line with more Bdnf-expressing cells following the TBI result from mature but not from newly generated neurons [54]. Thus, our results confirm the therapeutic potential of Bdnf for mild TBI. Since the application of exogenous Bdnf is limited as a result of the short half-life and difficulties in crossing the blood–brain barrier in humans, Bdnf mimetics seem to be promising candidates for promoting the Bdnf effects in TBI therapy. In our recent study, TrkB agonist 7,8-DHF, which is considered as a Bdnf mimetic, prevented the long-term adverse effects of a mild TBI in juvenile rats [58].

Neuroinflammatory response is a core process of TBI that tightly interacts with neuroreparative mechanisms. Resident immune cells of the brain, microglia, produce neurotrophic factors and regulate synaptic contacts providing neuronal plasticity and activity [59,60]. However, when activated, microglial cells release pro-inflammatory cytokines, free radicals, fatty acid metabolites, and quinolic acid along with depriving neurons of their regulatory control, which impairs learning and memory [61]. Although activated microglia protect the brain by removing cellular debris and toxic factors, uncontrolled activation of microglia is neurotoxic [62]. Nevertheless, inflammation is required and sufficient to induce the proliferation of neural progenitors and subsequent neurogenesis by activating injury-induced molecular programs that can be observed 24–48 h after TBI [63]. Here, the levels of a marker of microglial activation, Iba1, markedly increased in TBI groups during all days of the experiment vs. the respective Sham groups with a peak on Day 1, although a gradual relative decrease in the Iba1 expression was found in the TBI groups compared to acute period of injury (Days 0 and 1). Thus, microglia activation appears to play an essential role in the proper response to primary injury at the acute period of TBI, as microglial cells remove cellular debris and eliminate different noxious factors as well as trigger regenerative processes. Nonetheless, an excessive and/or prolonged microglia activation mediate further damage to the brain rather than neurorepair, and thus microglia modulation may represent a prospective approach in TBI therapy.

Another molecular target related to TBI in the present study was Hif1a, whose expression is initiated by TBI through the activation of the neuroinflammatory cascade, namely, cytokines and growth factors (IL-1, TNFα, TGFβ, insulin, insulin-like growth factor) via the PI3K/mTOR-dependent pathway [64]. Furthermore, local tissue hypoxia in the area of injury stops the polyhydroxylation and acetylation of Hif1a by the polyhydrolase, suppressing Hif1a ubiquitylation and its degradation in proteasomes [64]. The effects of Hif1a are dual, promoting both neuroprotection and aggravation of secondary TBI.

On the one hand, Hif1a induces neuroinflammatory responses including NLRP3-mediated microglial activation, pyroptosis, release of pro-inflammatory cytokines [65], as well as an increase in blood–brain barrier permeability and cerebral edema through the upregulation of LRRK2 expression [66]. On the other hand, in rodents, hippocampal trauma-induced neurogenesis involves increased expression of Hif1a [67]. Hif1a activates the expression of endothelial growth factor (Vegf), which has a wide neuroprotective potential, via the MAPK/CREB signaling pathway [67]. In addition to Vegf, Hif1a upregulates the expression of glucose transporters (GLUT 1 and 3), providing cells with a substrate for anaerobic glycolysis under conditions of local traumatic ischemia [56].

In the present study, the expression patterns of Hif1a in the telencephalon of laser-lesioned zebrafish were similar to those of Iba1. Hif1a expression increased right on the day of traumatic injury and stayed augmented until Day 7 after the lesion. The similarity was further confirmed by the strong positive correlation between the two biomarkers, which also correlated negatively with the NeuN expression. In general, our study supports the essential role of Hif1a in the neuroinflammatory response to mild TBI. Due to its multipotent properties, it does not seem correct to target Hif1a directly, while combinations attenuating neuroinflammation and enhancing the neurorepair through the Vegf and/or glucose transporters might be recommended at later stages of TBI to avoid secondary damage and provide neuroregeneration.

## 5. Conclusions

Our results suggest that the developed novel unique model of laser-induced focused non-penetrating TBI in zebrafish recapitulates mild TBI. The laser radiation produces a closed-head brain-specific injury suitable for further advanced and region-specific analyses. The advantages of this approach are that laser radiation allows us to localize injury and to vary the damage to nervous tissue by altering the laser power or duration of exposure to the laser irradiation. This model of neurotrauma was able to reproduce behavioral impairments strikingly, recapitulating in fish the responses observed in mammalian TBI models or in clinical practice and thus indicating face validity of the model. The results demonstrated a satisfactory degree of heterogeneity in the animals’ responses to the experimental exposure. Hence, this experimental animal model can be a useful tool to elucidate neurobiological mechanisms underlying the effects of laser-induced TBI. Molecular alterations found in the brain of laser-injured zebrafish are consistent with current notions about the pathogenesis of TBI, which evidences construct validity of the model. The results suggest that Bdnf promotes the survival and functional recovery of mature neurons in the lesioned area as a core event at the acute period of laser-induced mild TBI and recovery after the primary injury. Overall, our results confirm the therapeutic potential of Bdnf or its mimetics for mild TBI and suggest the modulation of neuroinflammatory response as another prospective approach in TBI therapy.

## Figures and Tables

**Figure 1 pharmaceutics-14-01751-f001:**
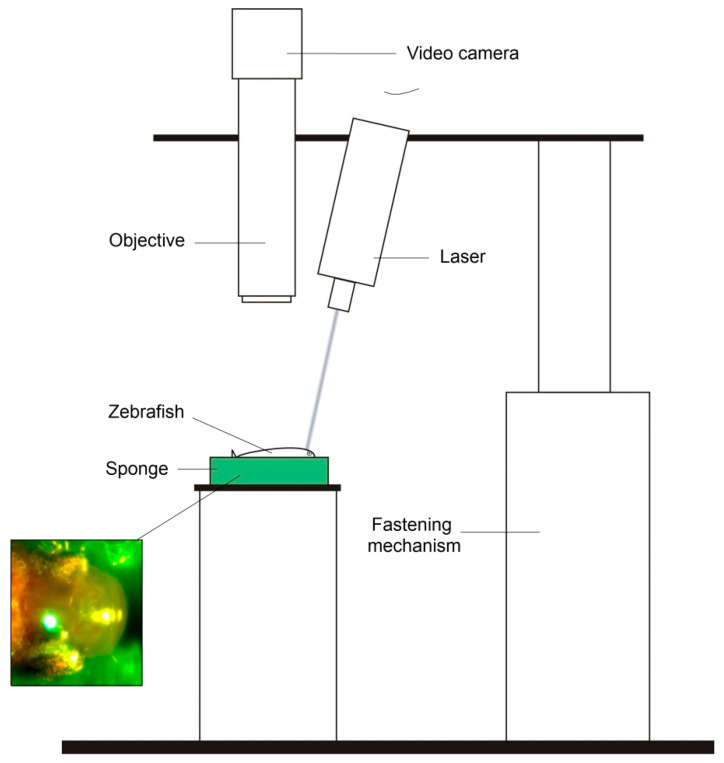
Laser set-up for precise targeted focused radiation system.

**Figure 2 pharmaceutics-14-01751-f002:**
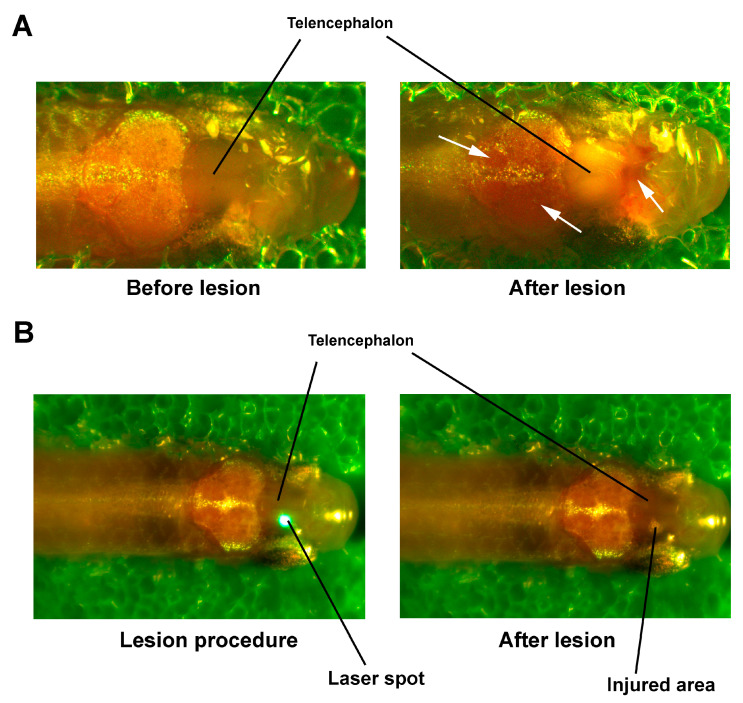
Laser-induced injury of the telencephalon in zebrafish. (**A**) Trauma of the whole telencephalon. White arrows show hemorrhages. (**B**) Local lesion of the right telencephalon.

**Figure 3 pharmaceutics-14-01751-f003:**
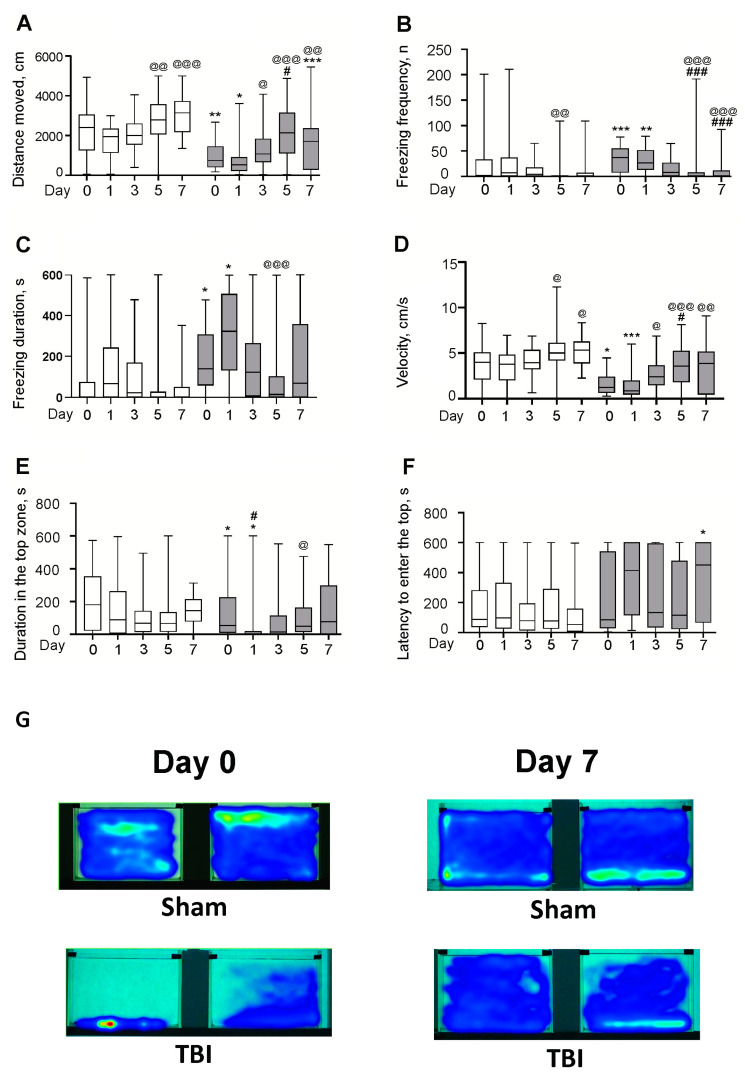
Effects of the laser-induced brain lesion (TBI model) on the behavior of zebrafish in the novel tank test on Days 0 (30 min after the lesion), 1, 3, 5, and 7 after the injury. General locomotion was evaluated by the distance traveled (**A**), the frequency (**B**) and the duration (**C**) of freezing, and by mean velocity (**D**). Cumulative duration in the top zone (**E**) and the latency to enter the top zone (**F**) served as indices of anxiety. White bars represent Sham groups; grey bars—TBI groups. Data are expressed as the median with first and third quartiles of the values obtained in an independent group of animals (*n* = 24–36 per group). * *p* < 0.05, ** *p* < 0.01, *** *p* < 0.001 vs. Sham group on the same day after the injury; # *p* < 0.05, ### *p* < 0.001 vs. TBI group on Day 0; @ *p* < 0.05, @@ *p* < 0.01, @@@ *p* < 0.001 vs. the same group (Sham or TBI) on Day 1. (**G**) Heatmap plots integrally reflecting the position in a tank of an animal during the test for two zebrafish from each group of TBI and Sham groups on Day 0 (30 min after injury) and Day 7.

**Figure 4 pharmaceutics-14-01751-f004:**
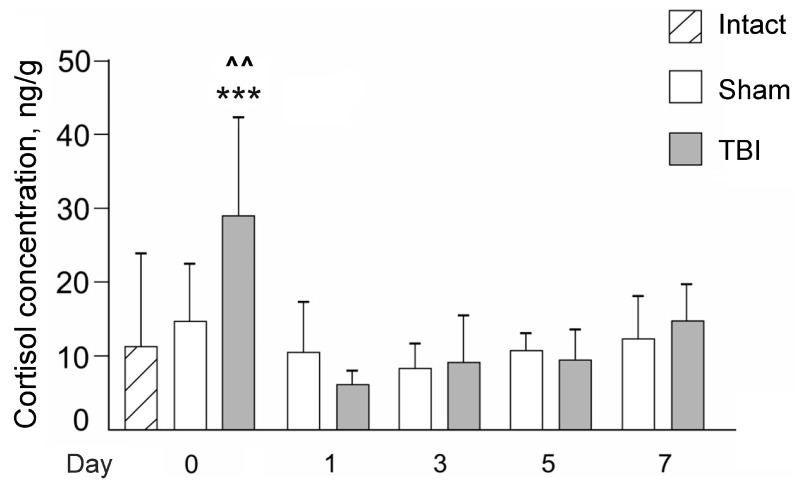
Effects of the laser-induced lesion (TBI model) on zebrafish whole-body cortisol on Day 0 (30 min after the lesion), 1, 3, 5, and 7 days after the injury. White bars represent Sham groups, grey bars—TBI groups; hatched bar—intact zebrafish. Data are expressed as mean ± SEM. *** *p* < 0.001 vs. Sham group on Day 0; ^^ *p* < 0.01 vs. Intact zebrafish on Day 0.

**Figure 5 pharmaceutics-14-01751-f005:**
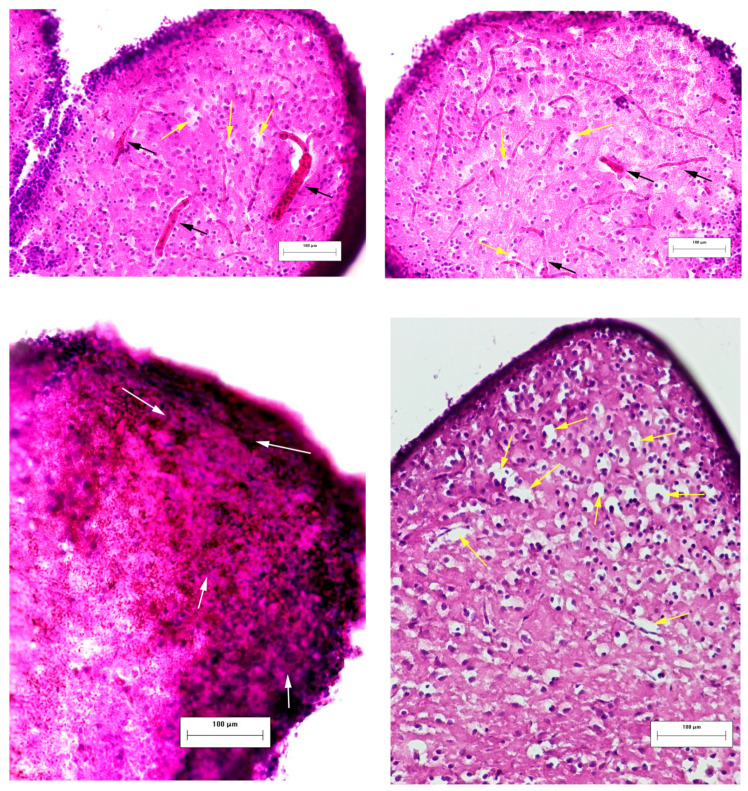
Effects of laser-induced lesion (TBI model) on the neuromorphology of the telencephalon one day after the injury in zebrafish. Representative images of brain cryosections of fish from the TBI group stained with hematoxylin and eosin. Black arrows show dilated vessels, white arrows—hemorrhage, yellow arrows—edema. Magnification, ×200.

**Figure 6 pharmaceutics-14-01751-f006:**
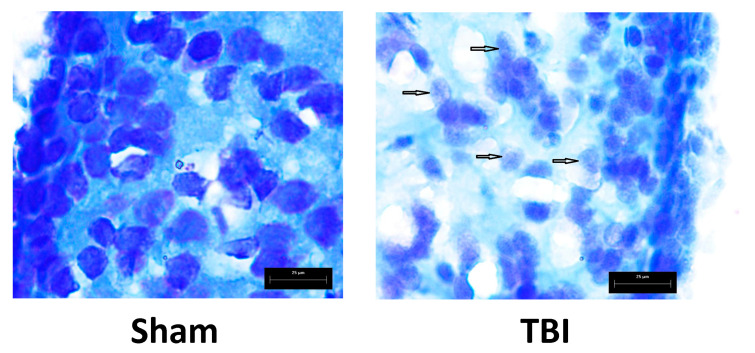
Effects of laser-induced lesion (TBI model) on neuromorphology of the telencephalon one day after the injury in zebrafish. Representative images of Nissl staining; arrows show degenerating neurons. Magnification, ×1000.

**Figure 7 pharmaceutics-14-01751-f007:**
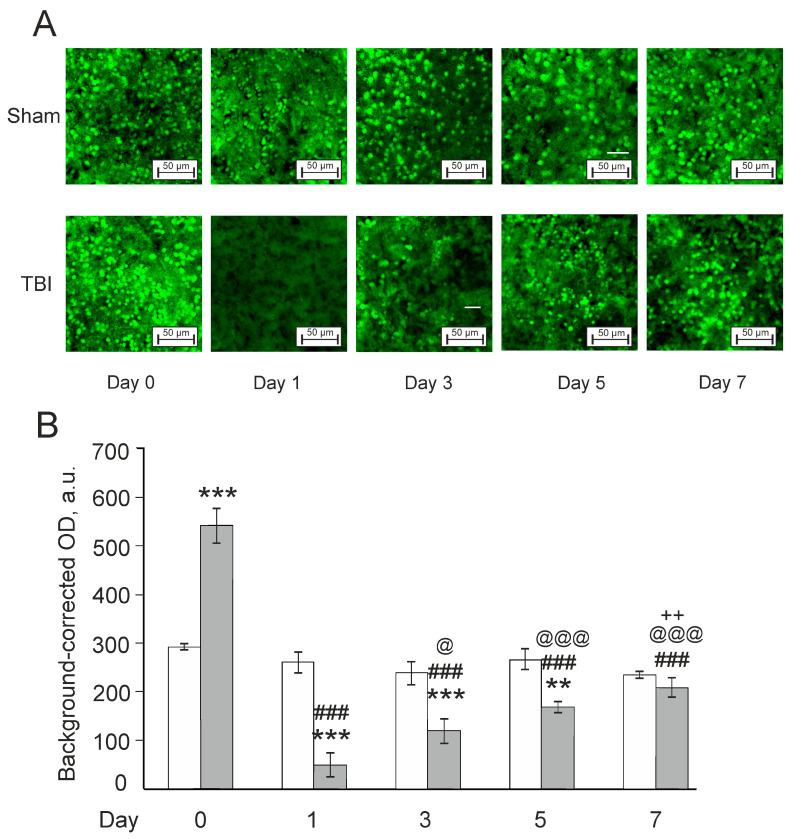
Effects of laser-induced lesion (TBI model) on the expression of NeuN in the telencephalon on Day 0 (30 min after the lesion), Days 1, 3, 5, and 7 after the injury in zebrafish. (**A**): Representative images of NeuN immunoreactivity in the telencephalon; magnification, × 200. (**B**): Quantitative results. White bars represent the Sham groups; grey bars—TBI groups. The data are expressed as mean ± SEM (*n* = 3–4 per group). ** *p* < 0.01, *** *p* < 0.001 vs. Sham group on the same day after the injury; ### *p* < 0.001 vs. TBI group on Day 0; @ *p* < 0.05, @@@ *p* < 0.001 vs. TBI group on Day 1; ++ *p* < 0.01 vs. TBI group on Day 3.

**Figure 8 pharmaceutics-14-01751-f008:**
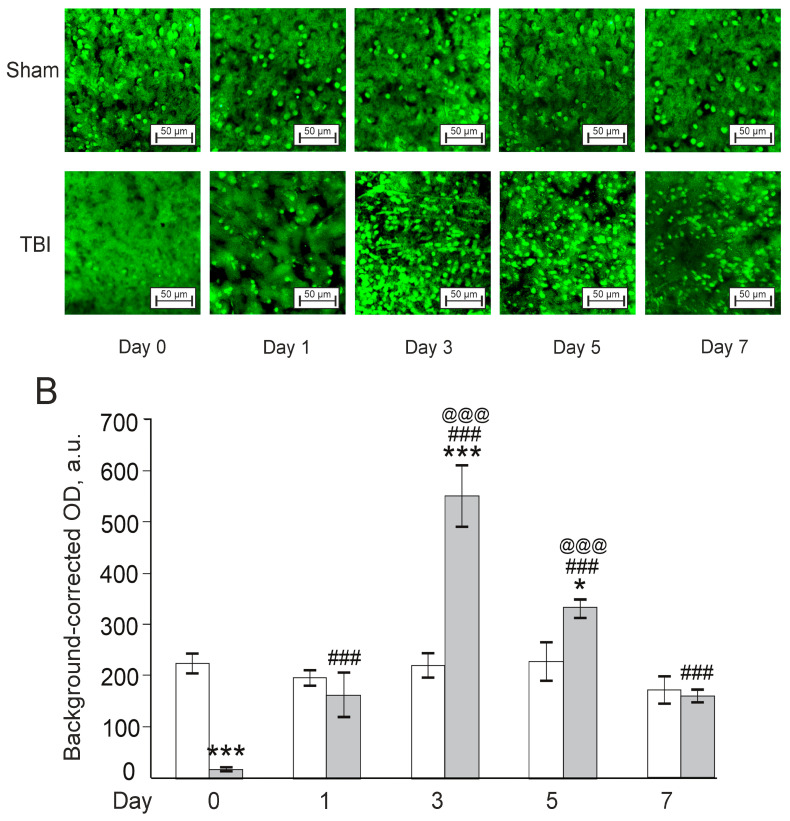
Effects of laser-induced lesion (TBI model) on the expression of Bdnf in zebrafish telencephalon on Day 0 (30 min after the lesion), Days 1, 3, 5, and 7 after the injury. (**A**): Representative images of Bdnf immunoreactivity in the telencephalon; magnification, ×200. (**B**): Quantitative results. White bars represent Sham groups, grey bars TBI groups. The data are expressed as mean ± SEM (*n* = 3–4 per group). * *p* < 0.05, *** *p* < 0.001 vs. Sham group on the same day after the injury; ### *p* < 0.001 vs. TBI group on Day 0; @@@ *p* < 0.001 vs. TBI group on Day 1.

**Figure 9 pharmaceutics-14-01751-f009:**
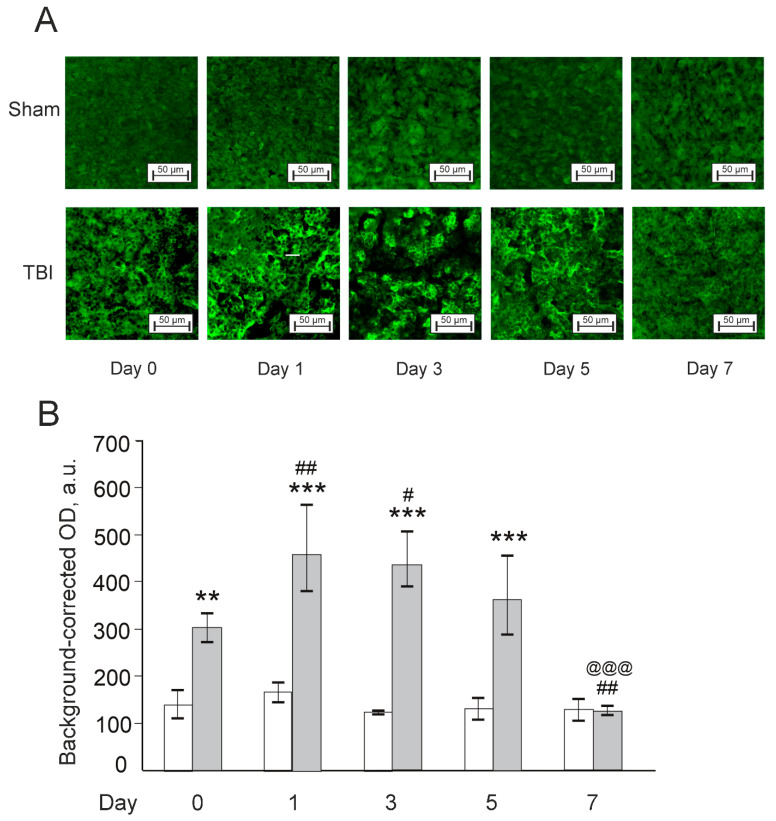
Effects of laser-induced lesion (TBI model) on the expression of Hif1a in zebrafish telencephalon on Day 0 (30 min after the lesion), Days 1, 3, 5, and 7 after the injury. (**A**): Representative images of Hif1a immunoreactivity in the telencephalon; magnification, ×200. (**B**): Quantitative results. White bars represent Sham groups, grey bars TBI groups. Data are expressed as mean ± SEM (*n* = 3–4 per group). ** *p* < 0.01, *** *p* < 0.001 vs. Sham group on the same day after the injury; # *p* < 0.05, ## *p* < 0.01 vs. TBI group on Day 0; @@@ *p* < 0.001 vs. TBI group on Day 1.

**Figure 10 pharmaceutics-14-01751-f010:**
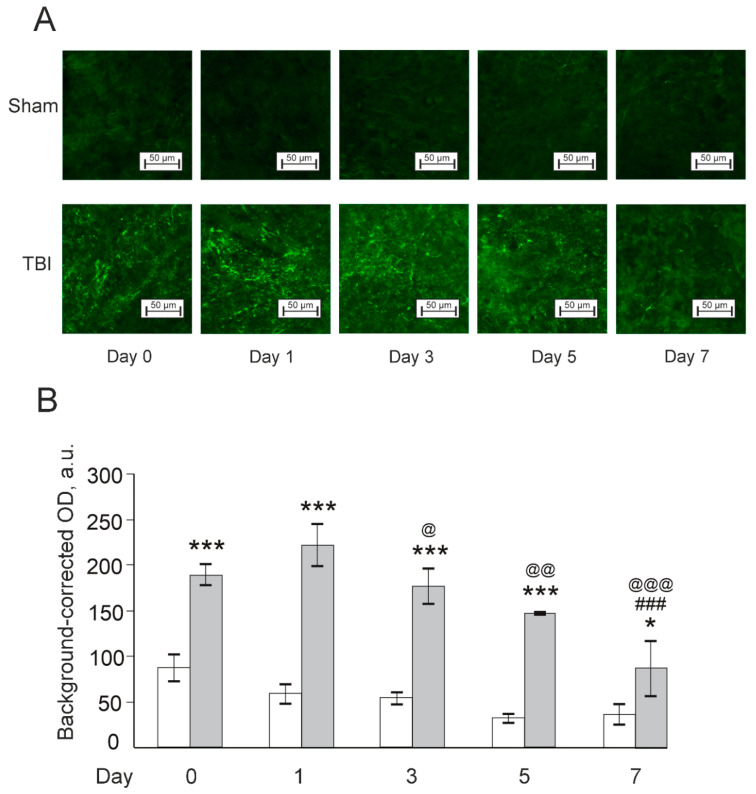
Effects of laser-induced lesion (TBI model) on the expression of microglial marker Iba1 in zebrafish telencephalon on Day 0 (30 min after the lesion), Days 1, 3, 5, and 7 after the injury. (**A**): Representative images of Iba1 immunoreactivity in the telencephalon; magnification, ×200. (**B**): Quantitative results. White bars represent Sham groups, grey bars TBI groups. Data are expressed as mean ± SEM (*n* = 3–4 per group). * *p* < 0.05, *** *p* < 0.001 vs. Sham group on the same day after the injury; ### *p* < 0.001 vs. TBI group on Day 0; @ *p* < 0.05, @@ *p* < 0.01, @@@ *p* < 0.001 vs. TBI group on Day 1.

## Data Availability

The data presented in this study are available upon a reasonable request from the corresponding authors.

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
