# Peer review of "A Novel Laser-Based Zebrafish Model for Studying Traumatic Brain Injury and Its Molecular Targets"

_pharmaceutics, 2022, doi:10.3390/pharmaceutics14081751_

Round 1
Reviewer 1 Report
In this manuscript, authors used the laser irradiation method to treat the zebrafish model for mimicking traumatic brain injury. Overall, it is a quite good research report. However, there are some comments and list as below:
1. Authors evaluated some specific neuronal relative biomarker molecules, but they did not mention why choosing them, such as Bdnf, Hif1a, Iba1, NenuN, but not others, such as IL-1, Vegf and so on. They should describe a rationale and at least some relative backgrounds.
2. In all figures, each panel should contain its own scale bar and should be clearly showed. In figure 4, should be pointed out where are the major observation areas via such as the arrows in figure 5, and its figure legend did not described clearly enough for each panel.
Page 5, Line 205, there is an uncompleted sentence containing only two words, “Locomotor activity”. Authors should carefully check what they want to present. Perhaps, to give it as a subtitle, such as “3.1.1. Locomotor activity”, that might be a better organization.
Author Response
Dear Reviewer, we would like to cordially thank you for the careful review of our manuscript. We have thoroughly revised the text considering all the comments, which helped us to improve the manuscript. All major corrections made in the text are highlighted with yellow color. We believe that the revised version would be more clear and interesting for the readership of the journal.
- Authors evaluated some specific neuronal relative biomarker molecules, but they did not mention why choosing them, such as Bdnf, Hif1a, Iba1, NeuN, but not others, such as IL-1, Vegf and so on. They should describe a rationale and at least some relative backgrounds.
The present study was aimed to develop and validate a novel model of non-invasive mild TBI in adult zebrafish, on the one hand, and to evaluate several potential molecular targets for neuroreparative treatment, on the other. Therefore, we focused on core processes of acute period of mild TBI, namely neurorepair and neuroinflammation. We chose well-validated biomarkers of neuronal injury and repair (NeuN and Bdnf) and neuroinflammation (marker of microglia activation Iba1). In addition, we assessed Hif1a. The latter attracted our attention because of its wide study in TBI models and dual effects, promoting both neuroprotection and aggravation of secondary TBI (inset, lines 86-90). We plan to analyze other molecular targets related to neuroinflammation including interleukin profiling or neurorepair (e.g., Vegf) in our future studies.
- In all figures, each panel should contain its own scale bar and should be clearly showed. In figure 4, should be pointed out where are the major observation areas via such as the arrows in figure 5, and its figure legend did not described clearly enough for each panel.
We revised the figure and its legend. We added scale bar to each image to clarify the calibration issue. Arrows of different colors were added to point out the major observation area.
Page 5, Line 205, there is an uncompleted sentence containing only two words, “Locomotor activity”. Authors should carefully check what they want to present. Perhaps, to give it as a subtitle, such as “3.1.1. Locomotor activity”, that might be a better organization.
Done. We reorganized the Section 3.1 by adding subsections 3.1.1.-3.1.5.
Reviewer 2 Report
In this work, the authors introduce a novel model of traumatic brain injury in zebrafish, evaluating the outcome of their procedure in terms of morphological and neurological changes in the fish. The authors conclude that their model reproduced the events related to mild TBI and therefore propose it as a novel preclinical model for the performance of neuroprotective drug screening.
This is an interesting and well-presented work introducing a potential very interesting model for screening of TBI treatments. A series of points should be clarified by the authors and some styling changes should be performed to help the understandable reading of the results:
- in page 4, paragraph 1, the authors mention that the laser injury cab ne immediately assessed by a change on the coloration of the tissue. An image showing this event would be highly illustrative at this point.
- Section 3.1 could be highly illustrative to add not just p values but actual distance travelled by fish in cm and also % of change (either from day to day or between groups). For example, when the authors mention ‘The novel tank test revealed significant differences in the distance traveled between Sham and TBI groups Days 0 (30 min after lesion; p<0.01), 1 (p<0.05)…’ they could write something like ‘The novel tank test revealed significant differences in the distance traveled between Sham and TBI groups Days 0 (30 min after lesion; Lsham= 2500 cm, LTBI= 800 cm, ΔL= -65%; p<0.01), 1 (Lsham= 2000 cm, LTBI= 400 cm, ΔL= -80%;p<0.05)…’
- same for the other measured parameters (freezing cumulative duration, velocity visiting the top zone and latency to enter the top.
- Y axis for plots C and E is the same in figure 2, please indicate somehow that plots E and F refer to the top area.
- in figure 2 G it is not clear what reflects the two top tanks and the two lower tanks, neither the two tanks of the left or the two tanks of the right for sham and for TBI (what are the 4 tanks in each group?
- could be illustrative to present heat maps for day 7, to show renormalization of behavior in TBI maps
- figure 4, please indicate to what corresponds each image (sham in the right and TBI on the left?, are all four images form TBI?, are different magnifications?
- In page 10 , at the end the authors state ‘ overall, the expression of neuronal marker NeuN was augmented in the acute period following TBI (30 min after lesion), but sharply dropped by Day 7. :
a) why NeuN is much higher in TBI group 30 min after lesion?. The explanation that may be massive release of protein from damaged nuclei is not totally convincing to me (looking at the image presented in figure 6 they do not look like protein released, it looks like cells). If authors find references in the literature to support this they should be provided, otherwise their comment is speculative and not supported by current results.
b) to me, NeuN levels at day 7 are slightly lower at day 7, I do not see the ‘sharp’ drop that the authors mention at day 7
- in page 17, paragraph 1. The authors state that ‘results suggest neuroregeneration of damaged neurons rather than repair due to neurogenesis’ this is confusing to me. The word neuroregeneration includes the generation of new neurons, neuroregeneration of damaged neuros seems more like neurorprotection, repair of damaged neurons, but not regeneration. Actually at the beginning of the next paragraph the authors state ‘ we consider than Bdnf promotes the survival and functional recovery of mature neurons’ please reformat.
-The discussion of novel neurons vs. repaired ones could be answered performing ki-67 staining, which is an indicator of cell proliferation and can be used to assess neurogenesis after injury. I understand that performing this staining right now it may be complicated and take excessive time, but I strongly suggest the authors to perform these studies to clarify the mechanisms of action (if not performed in this paper, at least mention as future studies)
- The conclusions section is too long a summary or results is not necessary, just what are the conclusions of the work.
Author Response
Dear Reviewer, we would like to cordially thank you for the careful review of our manuscript. We greatly appreciate your high esteem of the proposed TBI model and its potential for screening of TBI treatments. We have thoroughly revised the text considering all the comments, which helped us to improve the manuscript. All major corrections made in the text are highlighted with yellow color. We believe that the revised version would be more clear and interesting for the readership of the journal.
- in page 4, paragraph 1, the authors mention that the laser injury can be immediately assessed by a change on the coloration of the tissue. An image showing this event would be highly illustrative at this point.
We added a figure illustrating this point (Figure 2A).
- Section 3.1 could be highly illustrative to add not just p values but actual distance travelled by fish in cm and also % of change (either from day to day or between groups). For example, when the authors mention ‘The novel tank test revealed significant differences in the distance traveled between Sham and TBI groups Days 0 (30 min after lesion; p<0.01), 1 (p<0.05)…’ they could write something like ‘The novel tank test revealed significant differences in the distance traveled between Sham and TBI groups Days 0 (30 min after lesion; Lsham= 2500 cm, LTBI= 800 cm, ΔL= -65%; p<0.01), 1 (Lsham= 2000 cm, LTBI= 400 cm, ΔL= -80%;p<0.05)…’
- same for the other measured parameters (freezing cumulative duration, velocity visiting the top zone and latency to enter the top.
We revised the results in Section 3.1 according to your suggestion.
- Y axis for plots C and E is the same in figure 2, please indicate somehow that plots E and F refer to the top area.
Done.
- in figure 2 G it is not clear what reflects the two top tanks and the two lower tanks, neither the two tanks of the left or the two tanks of the right for sham and for TBI (what are the 4 tanks in each group?
- could be illustrative to present heat maps for day 7, to show renormalization of behavior in TBI maps
We revised the figure according to your suggestions. Heat maps for Day 7 were added. Four heat maps in previous version referred to four different fish of each group. In current version, we presented by two heat maps referred to two individual zebrafish of each group on each day.
- figure 4, please indicate to what corresponds each image (sham in the right and TBI on the left?, are all four images form TBI?, are different magnifications?
We revised the figure and its legend. All images correspond to TBI group and were made at the same magnification (200 x). We added scale bar to each image to clarify the calibration issue.
- In page 10 , at the end the authors state ‘ overall, the expression of neuronal marker NeuN was augmented in the acute period following TBI (30 min after lesion), but sharply dropped by Day 7. :
- a) why NeuN is much higher in TBI group 30 min after lesion?. The explanation that may be massive release of protein from damaged nuclei is not totally convincing to me (looking at the image presented in figure 6 they do not look like protein released, it looks like cells). If authors find references in the literature to support this they should be provided, otherwise their comment is speculative and not supported by current results.
Unfortunately, we cannot give definite answer to this question. NeuN is regarded as a specific neuronal marker of postmitotic neurons. This protein plays a role in neurospecific alternative splicing. The differences in the expression of this protein in cells are associated with both the constitutive characteristics of a neuron and its functional state. Most of the studies of pathological changes in neuronal populations indicate a weakening or disappearance of NeuN immunoreactivity in neurons. We have not found references about upregulation of NeuN. Moreover, 30 min after injury is too short period for huge production of NeuN protein de novo that could explain almost two-fold increase in the NeuN immunofluorescence. NeuN protein is predominantly associated with cell nuclei and, to a lesser extent, with the perinuclear cytoplasm. Most of the intranuclear NeuN is bound to the nuclear matrix. Hence, we hypothesize that laser irradiation disturbed that bound and NeuN was released to neuron cytoplasm. Nevertheless, this issue needs further detailed study while our suggestion should be proven. We adjusted this point in the manuscript (lines 521-532).
- b) to me, NeuN levels at day 7 are slightly lower at day 7, I do not see the ‘sharp’ drop that the authors mention at day 7
We revised the text according to your comment (lines 354-355).
- in page 17, paragraph 1. The authors state that ‘results suggest neuroregeneration of damaged neurons rather than repair due to neurogenesis’ this is confusing to me. The word neuroregeneration includes the generation of new neurons, neuroregeneration of damaged neuros seems more like neurorprotection, repair of damaged neurons, but not regeneration. Actually at the beginning of the next paragraph the authors state ‘ we consider than Bdnf promotes the survival and functional recovery of mature neurons’ please reformat.
Done.
-The discussion of novel neurons vs. repaired ones could be answered performing ki-67 staining, which is an indicator of cell proliferation and can be used to assess neurogenesis after injury. I understand that performing this staining right now it may be complicated and take excessive time, but I strongly suggest the authors to perform these studies to clarify the mechanisms of action (if not performed in this paper, at least mention as future studies)
The levels of NeuN expression had been markedly reduced until the Day 7 when the parameter in the TBI group was completely recovered. The period of a week is too short for neuronal precursor cells migration from the zones of proliferation to the injury site and differentiating into mature neurons. Nevertheless, we cannot exclude that laser irradiation may affect the process of neurogenesis and we plan to address this issue in our future studies.
- The conclusions section is too long a summary or results is not necessary, just what are the conclusions of the work.
We have shortened the text in the Conclusions.